# Normalizing Flows for Conformal Regression

**Nicolo Colombo**[1]

[1]`nicolo.colombo@rhul.ac.uk`,
Computer Science Department, Royal Holloway, University of London, Egham, Surrey, UK

## Abstract

Conformal Prediction (CP) algorithms estimate the uncertainty of a prediction model by calibrating its outputs on labeled data. The same calibration scheme usually applies to any model and data without modifications. The obtained prediction intervals are valid by construction but could be inefficient, i.e. unnecessarily big, if the prediction errors are not uniformly distributed over the input space. We present a general scheme to localize the intervals by training the calibration process. The standard prediction error is replaced by an optimized distance metric that depends explicitly on the object attributes. Learning the optimal metric is equivalent to training a Normalizing Flow that acts on the joint distribution of the errors and the inputs. Unlike the Error Reweighting CP algorithm of Papadopoulos et al. [2008], the framework allows estimating the gap between nominal and empirical conditional validity. The approach is compatible with existing locally-adaptive CP strategies based on reweighting the calibration samples and applies to any point-prediction model without retraining.

## 1 INTRODUCTION

In natural sciences, calibration often refers to comparing measurements of the same quantity made by a new device and a reference instrument.[1] In data science, calibrating

---

[1]The International Bureau of Weights and Measurements defines calibration as the *"operation that, under specified conditions, in a first step, establishes a relation between the quantity values with measurement uncertainties provided by measurement standards and corresponding indications with associated measurement uncertainties (of the calibrated instrument or secondary standard) and, in a second step, uses this information to establish a relation for obtaining a measurement result from an indication."*

a model means quantifying the uncertainty of its predictions. Parametric and non-parametric methods for model calibration have been proposed in the past. Examples of trainable post hoc approaches are Platt scaling [Platt et al., 1999], Isotonic regression [Zadrozny and Elkan, 2002], and Bayesian Binning [Naeini et al., 2015]. Here we focus on regression problems, where data objects have an attribute, $X \in \mathcal{X}$, and a real-valued label, $Y \in \mathbb{R}$. The model is a *point-like* predictor of the most likely label given its attribute, i.e. $f(X) \approx \mathrm{E}(Y|X)$. Calibrating $f$ would promote $f(X)$ to a Prediction Interval (PI), i.e. a *subset of the label space*, $C \subseteq \mathbb{R}$, that contains the unknown label, $Y$, with lower-bounded probability. Given a target *confidence level*, $1 - \alpha \in (0, 1)$, $C$ is *valid* if it contains the unknown label with probability at least $1 - \alpha$, i.e. if $\mathrm{Prob}(Y \in C) \geq 1 - \alpha$.

Conformal Prediction (CP) is a frequentist approach for producing valid PIs without making assumptions on the data-generating distribution, $P_{XY}$, or the prediction model, $f$ [Vovk et al., 2005, Shafer and Vovk, 2008]. PIs are obtained by evaluating the *conformity* between the predictions and the labels of a *calibration set*. The evaluation is based on a *conformity function*, e.g. the absolute residual, $a(Y, f(X)) = |Y - f(X)|$. Validity is guaranteed automatically by the properties of finite-sample empirical distributions. Different conformity functions, however, may produce non-equivalent PIs. Several criteria have been proposed to assess their *efficiency* [Vovk et al., 2016]. For real-valued labels, a straightforward criterion is the average size, $\mathrm{E}(|C|)$. If the model performs uniformly over the support of $P_{XY}$, the PIs obtained using $a = |Y - f(X)|$ have minimal average size. If the data are heteroscedastic, input-adaptive techniques may increase the PI efficiency because their size changes according to the performance of $f$, e.g. the prediction band shrinks where $|Y - f(X)|$ is small and grows where $|Y - f(X)|$ is large.

## 1.1 OUTLINE

We obtain *input-adaptive* PIs by learning calibration functions that depend on the object attributes explicitly. For simplicity, we assume the calibration samples, $(X_1, Y_1), \ldots, (X_N, Y_N)$, and the test object, $(X_{N+1}, Y_{N+1})$, are independently drawn from the same *joint distribution*, i.e. $(X_n, Y_n) \sim P_{XY}$. The method applies with minor changes if the samples are only *exchangeable* [Vovk et al., 2005]. Given a conformity function, $a : \mathbb{R}^2 \to \mathbb{R}$, and a confidence level, $1 - \alpha \in (0, 1)$, CP consists of two main steps,

1. computing the $(1 - \alpha)$-th *sample quantile*, $Q_A$, of the calibration scores, $A_n = a(Y_n, f(X_n))$, $n = 1, \ldots, N$, and

2. accept all possible test labels, $y \in \mathbb{R}$, for which $a(y, f(X_{N+1}))$ is smaller than $Q_A$.

If $A_n = |Y_n - f(X_n)|$, the PI at $X_{N+1}$ is the interval $C_A = [f(X_{N+1}) - Q_A, f(X_{N+1}) + Q_A] \subseteq \mathbb{R}$. Since $Q_A$ is the $(1 - \alpha)$-th sample quantile of $\{A_n\}_{n=1}^N$ and calibration and test samples are i.i.d., $C_A = \{y \in \mathbb{R}, a(y, f(X_{N+1})) \leq Q_A\}$ guarantees $\text{Prob}(Y_{N+1} \in C_A) \geq 1 - \alpha$. We say that $C_A$ is *marginally valid* because $Q_A$ approximates the quantile of the marginal distribution $P_A = \sum_{XY} P_{AXY}$[2]. In particular, there is no conditioning on the test input, $X_{N+1}$ [Vovk, 2012]. PIs with input-conditional coverage, $\text{Prob}(Y_{N+1} \in C_A | X_{N+1})$ *cannot* be obtained with finite data and without certain regularity assumptions on the data distribution [Lei and Wasserman, 2012, Vovk, 2012, Foygel Barber et al., 2021]. Approximating distribution-free conditionally-valid PIs is the goal of an active research stream (see Section 4). Existing methods are mostly based on importance-sampling techniques that temporarily break the data exchangeability [Lin et al., 2021, Tibshirani et al., 2019, Guan, 2023][3]. Our strategy is to preserve exchangeability at all times but change the definition of the conformity function, i.e. to replace $a$ with $b = b(a(Y, f(X)), X) \in \mathcal{B}$ and apply $b$ unconditionally to the calibration and test samples. Data exchangeability holds automatically provided $b$ is trained on a separate set. As in standard CP, we use the *transformed calibration samples*, $B_n = b(a(Y_n, f(X_n)), X_n), n = 1, \ldots N$, to compute a $B$-*space threshold*, $Q_B$, and build PIs that are marginally valid, i.e. of *constant* size, over $\mathcal{B}$. Local adaptability arises when

---

[2]Technically, marginal validity depends on the joint distribution of the calibration and test samples, i.e. $\text{Prob}(Y_{N+1} \in C_A) = P_{X_{N+1}Y_{N+1}X_1Y_1\ldots X_NY_N}(Y_{N+1} \in C_A)$.

[3]In Lin et al. [2021], Tibshirani et al. [2019], Guan [2023], the sample quantile of $\{A_n\}_{n=1}^N$ is replaced by the quantile of an importance-sampling estimate of the empirical input-conditional distribution, $P_{A|X} \approx \sum_{n=1}^N w_n(X)\mathbf{1}(A = A_n)$, where $\sum_{n=1}^N w_n(X) = 1$ and $w_n(X)$ depends on $X$ through a predefined function.

the PIs, $C_B = \{y \in \mathbb{R}, b(a(y, f(X_{N+1})), X_{N+1}) \leq Q_B\}$, are *mapped back* to the label space (by inverting $b$).

This work addresses the following problem,

*What transformations of the conformity function improve CP adaptivity? How can we optimize a transformation using a separate training set (from the same task)?*

We start by interpreting $b$ as a *Normalizing Flow* (NF), i.e. a coordinate transformation that maps a *source* distribution, $P$, into a *target* distribution, $P'$ [Papamakarios et al., 2021]. In our case, the source distribution is the joint distribution of the conformity scores and the object attributes, $P_{AX}$. The target is a factorized distribution, $P_{BX} = U_B P_X$, where $U_B$ is an arbitrary univariate distribution. In the $B$-space, the PIs are marginally valid and have constant size by construction. Maximal efficiency is guaranteed because the joint distribution factorizes, which implies $P_{B|X} = U_B$ for all $X$ and the equivalence between marginally and conditionally valid PIs. The practical problem is to enforce the factorization given the available data. The idea is to train $b$ by maximizing the likelihood of the transformed samples under $U_B$. When $b$ is invertible (in its first argument and for any $X_{N+1}$), the PIs are $C_B = \{y \in \mathbb{R}, a(y, f(X_{N+1})) \leq \xi_X\}$, $\xi_X = b^{-1}(Q_B, X_{N+1})$, with $b^{-1}$ defined by $b^{-1}(b(A, X), X) = A$. Intuitively, this produces locally adaptive PIs because $\xi_X$ approximates the unavailable conditional quantile $Q_{A|X}$. The approximation error depends on the distribution distance between the source and the target distributions, $P_{b(A,X),X}$ and $U_B P_X$.

## 1.2 AN EXAMPLE

Let $P_X = \text{Uniform}(\mathcal{X})$ be the uniform distribution over $\mathcal{X} = [0, 1]$ and $(X_1, Y_1), \ldots, (X_N, Y_N), (X_{N+1}, Y_{N+1}) \in \mathcal{X} \times \mathbb{R}$ a collection of i.i.d. random variables from

$$P_{XY} = P_{Y|X} P_X, \qquad (1)$$
$$P_{Y|X} \sim (\mathbf{1}_{<0.5} + \xi \, \mathbf{1}_{>0.5}) \mathcal{N}(0, 1)$$

where $\mathbf{1}_{<\frac{1}{2}} = \mathbf{1}(X < 0.5)$, $\mathbf{1}_{>0.5} = \mathbf{1}(X > 0.5)$, and $\xi = 5$. Assume we have the best-possible prediction model, i.e. $f(X) = \mathbb{E}(Y|X) = 0$, for any $X \in \mathcal{X}$. Let $a(Y, f(X)) = |Y - f(X)| = |Y|$ be the conformity measure and $A_n = |Y_n|$ the corresponding conformity scores, $n = 1, \ldots N + 1$. Choose a target confidence level, $1 - \alpha \in (0, 1)$, and let $Q_A$ be the $(1 - \alpha)$-th sample quantile of $\{A_n\}_{n=1}^N$, i.e. its $m_*$-th smallest element, $m_* = \lceil (1 - \alpha)(N + 1) \rceil$. If $N = 100$ and $\alpha = 0.05$, we have $m_* = 96$. The conformity scores, $A_1, \ldots, A_{N+1}$, are i.i.d. random variables because $(X_n, Y_n)$ are i.i.d. For any $X_{N+1}$, the marginal PI is $C_A = [f(X_{N+1}) - Q_A, f(X_{N+1}) + Q_A] = [-Q_A, Q_A]$, i.e. $C_A$ has the same width over the entire input space $\mathcal{X} = [0, 1]$.

Constant uncertainty does not correspond to the true model's prediction error (see Figure 1). The data are het-

eroscedastic because $P_{Y|X}$ in (1) depends on $X$ explicitly. As the calibration samples and $(X_{N+1}, Y_{N+1})$ are all drawn from $P_{XY} = P_{Y|X}\text{Uniform}([0,1])$, the test score, $a(Y_{N+1}, f(X_{N+1}))$, is smaller than $A_{m*}$ with probability $\frac{m*}{N+1}$, i.e. $\text{Prob}(Y_{N+1} \in C_A) = \frac{m*}{N+1}$. Figure 1 shows that constant-size marginal PIs are valid but *inefficient*. In particular, $C_A$ is too large when $X_{N+1} < 0.5$ and too small when $X_{N+1} > 0.5$. An adaptive CP algorithm should output PIs that are smaller or larger than $C_A$ when $X_{N+1} < 0.5$ or $X_{N+1} > 0.5$.

We aim to learn a *locally adaptive conformity functions*, $b = b(A, X)$, that produces these adaptive PIs automatically, i.e. without partitioning the input space and using the standard CP procedure described in Section 1.1. Let $B_n = b(A_n, X_n) \in \mathcal{B}$ and $Q_B$ be the sample quantile of $\{B_n\}_{n=1}^N$. In $\mathcal{B}$, PIs are defined as in Section 1.1, i.e. $C_B = \{y \in \mathbb{R}, b(|y|, X_{N+1}) \leq Q_B\}$ and have constant size (see Figure 2). Assuming $b$ is monotonic in $A_n$, there exist $b^{-1}$ such that $b^{-1}(b(A, X), X) = A$. The inverse transformation, $b^{-1}$, can be used to map $C_B$ back to the label space, i.e. to rewrite the PIs as $\{y \in \mathbb{R}, |y| \leq b^{-1}(Q_B, X_{N+1})\}$. For improving PI efficiency, we need a $b$ such that $b^{-1}(Q_B, X_{N+1})$ is smaller than $Q_A$ for $X_{N+1} < 0.5$ and larger than $Q_A$ for $X_{N+1} > 0.5$.

Similar to the Mondrian CP algorithm [Vovk et al., 2005], we split $\{(A_n, X_n)\}_{n=1}^N$ into $D_{<0.5} = \{(A_n, X_n), X_n < 0.5\}_{n=1}^N$ and $D_{>0.5} = \{(A_n, X_n), X_n > 0.5\}_{n=1}^N$. Since $Y_n|X_n \sim \mathcal{N}(0,1)$ for all $(Y_n, X_n) \in D_{<0.5}$ and $Y_n|X_n \sim \mathcal{N}(0,5)$ for all $X_n \in D_{>0.5}$, the quantile of $P_{A|X}$ can be written as $Q_{A|X} = \mathbf{1}_{<0.5}Q_{A|X<0.5} + \mathbf{1}_{>0.5}Q_{A|X>0.5}$, where $Q_{A|X<0.5}$ is the $m_*$-th smallest elements of $D_{<0.5}$, $m_* = \lceil(1-\alpha)(|D_{>0.5}| + 1)\rceil$ (idem for $X < 0.5$). As expected, the conditional quantile depends on $X$ through $\mathbf{1}_{<0.5} = \mathbf{1}(X < 0.5)$ and $\mathbf{1}_{>0.5} = \mathbf{1}(X > 0.5)$. This implies the conditionally-valid PIs, $C_{A|X} = [-Q_{A|X}, Q_{A|X}]$, will depend on the location of the test object $X_{N+1}$, i.e. on whether $X_{N+1} < 0.5$ or $X < 0.5$. In this special case, the conditionally valid PIs for $X < 0.5$ and $X > 0.5$ are equivalent to the marginal PIs of the regions $[0, 0.5]$ and $[0.5, 1]$. Partitioning the calibration data is optimal if i) the sample size is large enough and ii) we know the data generating distribution. Otherwise, we need a more general approach.

Let

$$b_{flow} = \log\left(\frac{A}{\gamma + |g(X)|^2}\right), \qquad (2)$$

$$g(X) = \theta_1 X + \theta_2 X^2 + \theta_3 X^3$$

where $\theta = (\theta_1, \theta_2, \theta_3) \in \mathbb{R}^3$ is a free parameter and $\gamma = 0.01$. For any $X$ and $\theta$, $b_{flow}(A, X)$ is a monotonic (and hence invertible) function of $A = |Y|$. Analogously, let $b_{ER} = \frac{A}{\gamma + |g(X)|^2}$ as in the Error Reweighted (ER) CP algorithm of Papadopoulos et al. [2008]. $b_{ER}$ is also a monotonic and invertible function of $A = |Y|$. Let $\{b_{flow}(A_n, X_n)\}_{n=1}^N$ and $\{b_{ER}(A_n, X_n)\}_{n=1}^N$ be the *trans-*

*formed calibration sets* obtained using $b_{flow}$ and $b_{ER}$. To compare our strategy and the ER algorithm, we look at the efficiency of $b_{flow}$ and $b_{ER}$ when $\theta_{flow}$ and $\theta_{ER}$ are trained through the proposed NF scheme or the error-fitting heuristic of Papadopoulos et al. [2008]. Figure 2 shows a sample of the original calibration scores $\{A_n\}_{n=1}^N$ and the transformed scores obtained through $b_{flow}$ and $b_{ER}$ when $\theta_{flow} \neq \theta_{ER}$ are optimized following the corresponding strategies on a separate training data set. [4]

Let $Q_B$ be the $(1-\alpha)$-th (marginal) sample quantile of $\{B_n = b_{flow}(A_n, X_n)\}_{n=1}^N$. By definition, $Q_B = \log\left(A_{n_*}(\gamma + |g(X_{n_*})|^2)^{-1}\right)$, for some $n_*$ such that $\lceil(1-\alpha)(N+1)\rceil$ elements of $\{B_n\}_{n=1}^N$ are smaller than or equal to $Q_B$. In general, since $b_{flow}$ depends on $X$ explicitly, this does not imply there are $\lceil(1-\alpha)(N+1)\rceil$ elements of $\{A_n\}_{n=1}^N$ smaller than or equal to $A_{n_*}$. The exchangeability of $B_n$ and $B_{N+1} = b_{flow}(|Y_{N+1}|, X_{N+1})$ guarantees the validity of the $B$-space PIs, i.e. $\text{Prob}(B_{N+1} \leq Q_B) = \frac{n_*}{N+1}$. The validity of the corresponding label-space PIs, $\text{Prob}\left(|Y_{N+1}| \leq e_B^Q(\gamma + |g(X_{N+1})|^2)\right) = \frac{n_*}{N+1}$, follows from the monotonicity of $b_{flow}$. [5] Similar arguments apply to $b_{ER}$.

The above holds for any $X_{N+1}$ and any $\theta$. We aim to choose a $\theta$ that improves the efficiency of $C_B$, e.g. reduces its average size. In Papadopoulos et al. [2008], $\theta$ would be tuned to make $g(X)$ a model of the conditional residuals, i.e. $\theta_{ER} = \arg\min_\theta \sum_{n'=1}^N |Y_n^2 - |g(X_n)|^2|^2$, where $\{(A_{n'}, X_{n'})\}_{n'=1}^N$ is a separate *calibration-training set* of labeled samples. In this work, we interpret $(A, X) \rightarrow (b_{flow}(A, X), X)$ as an NF acting on the joint distribution of the conformity scores and the inputs, $(A, X) \sim P_{AX}$. $b_{flow}$ is then trained by maximizing the likelihood of $(B, X)$ under a target factorized distribution, $P_{BX} = U_B P_X$. If the NF transforms $(A, X)$ into $(B, X) \sim P_{BX}$ exactly, the obtained marginally-valid $B$-space PIs have maximal efficiency because $P_{B|X} = U_B = \sum_X P_{BX}$ for all $X$. The choice of $U_B$ is arbitrary, provided its support is compatible with the transformation class, e.g. choosing $U_B = \text{Uniform}([0,1])$ would not be ideal in this case because $b_{flow} \in \mathbb{R}$. We choose $U_B = \mathcal{N}(0,1)$ instead and let

$$\theta_{flow} = \arg\min_\theta \sum_{n'=1}^N |b_{flow}(A_{n'}, X_{n'})|^2 \qquad (3)$$

where we use $u_B \propto \exp^{-\frac{B^2}{2}}$ and can drop the transformation Jacobian, $\partial_A b_{flow} = \frac{1}{A}$, because it does not depend

---

[4]As $\log(t)$ is a monotonic and input-independent transformation, the PIs obtained from $b_{ER}$ and $b_{flow} = \log \circ b_{ER}$ are equivalent if we use the same localization function, e.g. if we set $\theta = \theta_{flow} = \theta_{ER}$.

[5]We use $\text{Prob}(B_{N+1} \leq Q_B) = \text{Prob}(A_{N+1} \leq b_{flow}^{-1}(Q_B, X_{N+1})) = \text{Prob}(|Y_{N+1}| \leq e_B^Q(\gamma + |g(X_{N+1})|^2)) = \text{Prob}(Y_{N+1} \in C_B)$.

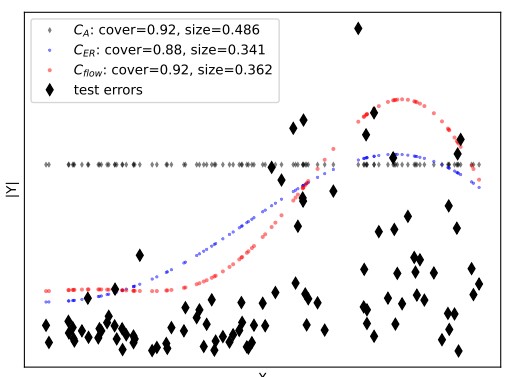

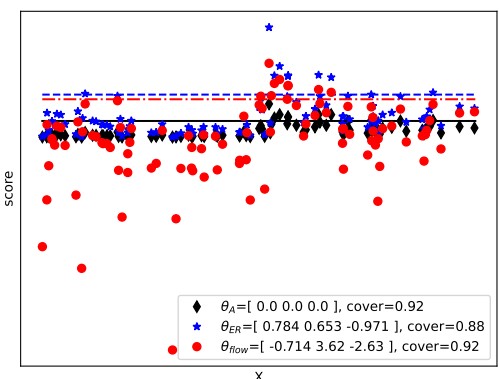

Figure 1: A test sample of conformity scores (black diamonds) and the upper bound of the marginal PIs (black dots) and the adaptive PIs obtained through the ER CP algorithm of Papadopoulos et al. [2008] (blue dots) and the NF approach (red dots). The nominal confidence level is $1 - \alpha = 0.9$ for all algorithms.

Figure 2: A calibration sample of the original conformity scores (black diamonds) and the scores obtained by transforming them with $b_{ER}$ (blue stars) and $b_{flow}$ (red dots). The solid and dashed lines represent the corresponding $(1 - \alpha)$-th sample quantiles, $1 - \alpha = 0.9$.

on $\theta$. As for $b_{ER}$, $\{(A_{n'}, X_{n'})\}_{n'=1}^N$ is a separate training set, which we will not use to calibrate or test the trained CP algorithms. Figure 1 shows the label-space PIs obtained by setting $\theta = \theta_{flow}$ in $b_{flow}$ ($C_{flow}$, in red) and $\theta = \theta_{ER}$ in $b_{ER}$ ($C_{ER}$, in blue).

$N^{-1} \sum_{n=1}^N \mathbf{1}(Z = Z_n)$, i.e.

$$Q_Z = \inf_q \{q \in \mathcal{Z}, \sum_{n=1}^N \mathbf{1}(Z_n \leq q) \geq n_*\} \qquad (5)$$

$$n_* = \lceil (N+1)(1-\alpha) \rceil$$

where $\lceil s \rceil$ the smallest integer greater than or equal to $s \in \mathbb{R}$. Assuming ties occur with probability 0, i.e. $\mathrm{Prob}(Z_n = Z_{n'}) = 0$ for any $n \neq n'$, $Q_Z$ is the $n_*$-th smallest element of $\{Z_n \sim P_Z\}_{n=1}^N$. CP validity is a direct consequence of

## 2  THEORY

In this section, $\mathcal{X}$ is an arbitrary attribute space and $\{(X_n, Y_n) \in \mathcal{X} \times \mathbb{R}\}_{n=1}^{N+1}$ a collection of i.i.d. random variables from an unknown joint distribution, $P_{XY} = P_{Y|X} P_X$. The regression model, $f(X_n) \approx \mathrm{E}(Y_n | X_n)$, $n = 1, \ldots, N+1$, is assumed to be pre-trained on separate data.

**Lemma 2.1 (Quantile Lemma Tibshirani et al. [2019])**
*Let $Z_1, \ldots, Z_N, Z_{N+1} \in \mathbb{R}$ be a collection of i.i.d. random variables and $Q_Z$ be the $(1 - \alpha)$-th sample quantile of $\{Z_n\}_{n=1}^N$ defined in (5). If ties occur with probability 0,*

$$\mathrm{Prob}\left(Z_{N+1} \leq Q_Z\right) = \frac{\lceil (1-\alpha)(N+1) \rceil}{N+1} \qquad (6)$$

The lemma first appeared in Papadopoulos et al. [2002]. Slightly different proofs can be found in Lei and Wasserman [2014], Tibshirani et al. [2019], Angelopoulos and Bates [2021]. The standard CP bounds, $1 - \alpha \leq \mathrm{Prob}(Z_{N+1} \leq Q_Z) \leq 1 - \alpha + \frac{1}{N+1}$, follows from $\lceil s \rceil - s \geq 0$ and $(1-\alpha)(N+1) \leq \lceil (1-\alpha)(N+1) \rceil \leq (1-\alpha)(N+1)+1$. Asymptotically, $Q_Z$ is normally distributed around $\bar{Q}_Z$ with variance $\sigma^2 = \frac{(1-\alpha)\alpha}{N p_Z(\bar{Q}_Z)}$, where $p_Z(\bar{Q}_Z)$ is the density of $P_Z$ evaluated at $Z = \bar{Q}_Z$, with $\bar{Q}_Z$ defined in (4).

### 2.1  QUANTILES

Given a random variable, $Z \in \mathcal{Z}$ and its distribution, $P_Z$, let $F_Z(z) = P_Z(Z \leq z)$ be the Cumulative Distribution Function of $P_Z$. The $(1 - \alpha)$-th quantile of $Z \sim P_Z$ is

$$\bar{Q}_Z = \inf_q \{q \in \mathcal{Z} : F_Z(q) \geq (1-\alpha)\} \qquad (4)$$

When $Z$ is continuous, $F_Z$ is strictly increasing and $\bar{Q}_Z = F_Z^{-1}(1-\alpha)$. The $(1 - \alpha)$-th sample quantile of a collection of i.i.d. random variables, $\{Z_n \sim P_Z\}_{n=1}^N$, is the $(1 - \alpha)$-th quantile of their empirical distribution $P_Z \approx$

### 2.2  CONFORMITY SCORES

A conformity score is a random variable, $A = a(f(X), Y)$, that describes the conformity between a prediction, $f(X)$,

and the corresponding label, $Y$. A standard choice is $a = |Y - f(X)|$. Let $P_{AX}$ be the distribution of the i.i.d. random variables $\{(A_n = |Y_n - f(X_n)|, X_n)\}_{n=1}^{N+1} =$. Lemma 2.1 guarantees the validity of the symmetric PI,

$$C_A = [f(X_{N+1}) - Q_A, f(X_{N+1}) + Q_A] \qquad (7)$$

when $Q_A$ is the $(1-\alpha)$-th sample quantile of $\{A_n\}_{n=1}^N$. We may also let the conformity scores be $B = b(A)$, where $b$ is a *global monotonic function* of its argument, e.g. $b(A) = -A^{-1}$ or $b(A) = \log A$. In that case, we obtain the PIs by inverting $b$ and letting $C_B = [f(X_{N+1}) - b^{-1}(Q_B), f(X_{N+1}) + b^{-1}(Q_B)]$, where $Q_B$ is the $(1-\alpha)$-th sample quantile of $\{B_n = b(A_n)\}_{n=1}^N$ and $b^{-1}$ is defined by $b^{-1} \circ b(A) = A$. For example, $b^{-1}(Q_B) = -\frac{1}{Q_B}$ if $B = -\frac{1}{A}$ and $b^{-1}(Q_B) = \exp(Q_B)$ if $B = \log A$. Assuming ties occur with probability 0, $Q_A$ is the $n_*$-th smallest element of $\{A_n\}_{n=1}^N$, with $n_* = \lceil (1-\alpha)(N+1) \rceil$. Let $A_*$ be that element. The $(1-\alpha)$-th sample quantile of the transformed scores, $Q_B$, is the $\lceil (1-\alpha)(N+1) \rceil$-th smallest element of $\{b(A_n)\}_{n=1}^N$. If $b$ is monotonic and applies globally to all samples, $b(A_n) < b(A_{n'})$ if and only if $A_n < A_{n'}$, for any $n \neq n'$. Then $Q_B = b(A_*)$ and $b^{-1}(Q_B) = Q_A$, i.e. the size of the PIs does not depend on $b$. If $b$ depends on the input, $b(A_n, X_n) < b(A_{n'}, X_{n'})$ does not imply $A_n < A_{n'}$, for any $n \neq n'$, i.e. the PIs depends on $b$.

## 2.3 NORMALIZING FLOWS

This work is about finding an input-dependent transformation $b = b(A, X)$ that changes the PIs to make them locally adaptive and more efficient automatically, i.e. without splitting the calibration data set and applying any existing CP algorithm. In what follows, we assume $b$ always satisfies

**Assumption 2.2** *For $\mathcal{A}, \mathcal{B} \subset \mathbb{R}$, $b : \mathcal{A} \times \mathcal{X} \to \mathcal{B}$*

1. *is strictly increasing on its first argument, i.e. $J_b(A, X) = \frac{\partial}{\partial A} b(A, X) > 0$ for all $(A, X)$ and*

2. *its domain and co-domain are the same for all $X \in \mathcal{X}$.*

Let $b^{-1}(B, X)$ be defined by $b^{-1}(b(A, X), X) = A$. The assumption on the domain and co-domain of $b$ guarantees $b^{-1}(b(A, X'), X)$ is well defined for any $X \neq X'$. We avoid over-fitting by letting $b$ be smooth in $X$ and $A$. Since $b$ acts on random variables and obeys Assumption 2.2, we can interpret it as (part of) an NF. Let $P_Z$ and $U_Z$ be two distributions with the same support, $\mathcal{Z}$. An NF is an invertible coordinate transformation from $\mathcal{Z}$ to $\mathcal{Z}$ such that

$$Z' = \phi_b(Z) \sim U_{Z'}, \quad Z = \phi_b^{-1}(Z') \sim P_Z \qquad (8)$$

In our case, $Z = (A, X)$, $Z' = (B, X)$, and $\phi_b(A, X) = (b(A, X), X)$. The Jacobian of $\phi_b$ is a $(|\mathcal{X}|+1)$-dimensional

squared matrix, $J_{\phi_b}$, such that $J_{\phi_b ij} = 0$ for all $i, j > 1$ and $i \neq j$, $J_{\phi_b ii} = 1$ for all $i > 1$, $J_{\phi_b 1i} = \frac{\partial}{\partial X_i} b(A, X)$ for all $i > 1$, and $J_{\phi_b 11} = \frac{\partial}{\partial A} b(A, X)$. We often use $J_b(A, X)$ instead of $J_{\phi_b 11}$. Assumption 2.2 implies $J_{\phi_b 11} > 0$ and guarantees the invertibility of $\phi_b$ because, for any $(A, X)$, $\det(J_{\phi_b}(A, X)) = \prod_{i=1}^{|\mathcal{X}|+1} J_{\phi_b ii}(A, X) = J_{\phi_b 11}(A, X)$ is strictly positive. When not explicitly required, we drop the trivial part of $\phi_b$ and use $b$ fo $\phi_b$ *and* $\phi_{b1}$ depending on the context. See Papamakarios et al. [2021] for a review of using NFs in inference tasks.

## 2.4 VALIDITY

Given an NF, $b$, we let the associated marginal PI at $X_{N+1}$ be

$$C_B = [f(X_{N+1}) - \delta, f(X_{N+1}) + \delta] \qquad (9)$$
$$\delta = b^{-1}(Q_B, X_{N+1})$$

where $Q_B$ is the $(1-\alpha)$-th sample quantile of $\{B_n = b(A_n, X_n)\}_{n=1}^N$. If ties occur with probability 0, the validity of $C_B$ defined in (9) is guaranteed by

**Lemma 2.3** *Let $b$ satisfy Assumption 2.2 and $C_B$ be the PI defined in (9). Then*

$$\mathrm{Prob}(Y_{N+1} \in C_B) = \frac{\lceil (1-\alpha)(N+1) \rceil}{N+1} \qquad (10)$$

The transformation is globally defined but *acts differently* on the samples, e.g. we may have $b(A, X_n) \neq b(A, X_{n'})$ for some $A \in \mathcal{A}$ and $n \neq n'$. The ranking of the *original scores*, $\{A_n\}_{n=1}^N$, may differ from the ranking of the *transformed scores*, $\{B_n\}_{n=1}^N$, i.e. $A_1 < A_2 < \cdots < A_N$ may not imply $B_1 < B_2 < \cdots < B_N$. This happens if $A_n < A_{n'}$ and $b(A_n, X_n) > b(A_{n'}, X_{n'})$ for some $n \neq n'$. While validity is automatically guaranteed because calibration and test samples remain exchangeable, we may have $C_A \neq C_B$, e.g. when $b$ changes the ranking of the calibration samples. Under further mild assumptions on $b$, Lemma 2.4 shows that we can find a test object for which the PIs obtained with $b$ and $a$ have difference sizes, i.e. $|C_B| \neq |C_A|$.

**Lemma 2.4** *Let $\{A_n\}_{n=1}^{N+1}$ be a collection of i.i.d. continuous random variables. Assume $b$ satisfies Assumption 2.2. Then, if $b(A_n, X_{N+1}) \neq b(A_n, X_n)$ for any $n = 1, \ldots, N$,*

$$|C_B| \neq |C_A| \qquad (11)$$

*with $C_B$ and $C_A$ defined in (9) and (7).*

## 2.5 EXACT NORMALIZING FLOWS

In some cases, marginally valid PIs are also conditionally valid for any $X_{N+1} \in \mathcal{X}$, i.e. $C_A$ defined in (7) obeys

$$\mathrm{Prob}(Y_{N+1} \in C_A | X_{N+1}) \geq 1 - \alpha \qquad (12)$$

This may occur when $P_{AX}$ has a specific form. When the data are *not* heteroscedastic, i.e. $P_{AX} = P_{A|X}P_X = P_A P_X$, the equivalence of marginal and conditional PIs is guaranteed by

**Theorem 2.5** *Let* $P_{AX} = P_A P_X$ *for any* $X \in \mathcal{X}$. *For any* $X_{N+1} \in \mathcal{X}$, $C_A$ *defined in* (7), *obeys*

$$\text{Prob}(Y_{N+1} \leq C_A | X_{N+1}) = \frac{\lceil (N+1)(1-\alpha) \rceil}{N+1} \quad (13)$$

Theorem 2.5 is a straightforward consequence of the Bayesian theorem and Lemma 2.1. We include it here because it suggests we can find an NF that localizes the PIs. The idea is to train $b$ to make $C_B = C_{b(A)}$ conditionally valid through Theorem 2.5, i.e. to make $b$ such that $(b(A_n, X_n), X_n) = (B_n, X_n) \sim P_{BX} = P_B P_X$. Interpreting $b$ as an NF, we can find a near-optimal $b$ through standard NF-training techniques, e.g. by *maximizing the likelihood of the transformed scores under an arbitrary target distribution, $U_B$, that does not depend on the input*. Given samples from $A$, we need the composition between the target distribution and the score transformation, $b$. In particular, $\int_x^{x'} dx p(f(x)) = \int_{f(x)}^{f(x')} \frac{dy}{f'(f^{-1}(y))} p(y)$ implies the density of the composition is $p(B, X) = u(b(A, X)) J_b(A, X)) p(X)$. The objective function is

$$\ell(b) = \text{E}\left(\log u(B) p(X)\right) \quad (14)$$
$$= \text{E}\left(\log\left(u(b(A, X)) |J_b(A, X)|\right)\right) + \ell_0$$

where $u$ is the density of the (arbitrary) target distribution, $U_B$, and $\ell_0 = \text{E}(\log p(X))$ does not depend on $b$. Fix a given target distribution, $U_B$, e.g. let $U_B$ be the univariate Gauss distribution or $U_B \sim \text{Uniform}([0, 1])$. Assume there exists an NF, $b$, that satisfies Assumption 2.2 and is such that $P_{BX} = U_B P_X$ for any $(A, X)$ when $B = b(A, X)$. Then, $C_B$ defined in (9) is conditionally valid at $X_{N+1}$. The claim is supported by

**Corollary 2.6** *Let* $U_B$ *be an arbitrary univariate distribution and $b$ an NF satisfying Assumption 2.2. If* $(B, X) = (b(A, X), X) \sim P_{BX} = U_B P_X$ *for any* $(A, X)$, $C_B$ *defined in* (9) *obeys*

$$\text{Prob}(Y_{N+1} \in C_B | X_{N+1}) = \frac{\lceil (1-\alpha)(N+1) \rceil}{N+1} \quad (15)$$

Corollary 2.6 follows from Lemma 2.3 and the monotonicity of $b$. There is no contradiction with the negative results of Lei and Wasserman [2012], Vovk [2012] because exact factorization can not be achieved with finite data.

## 2.6 NON-EXACT NORMALIZING FLOWS

Let $\hat{b}$ be an NF trained by maximizing a finite-sample empirical estimation of the likelihood defined in (14). We do not expect $\hat{b}$ to factorize $P_{BX}$ exactly but assume it approximates the ideal optimal transformation, $b$, defined in Corollary 2.6 in the Huber sense. More precisely, we let $\epsilon > 0$ quantify the discrepancy between the two transformations and

$$\hat{b} = (1-\epsilon)b + \epsilon\delta, \quad (16)$$

where $\delta = \delta(A, X)$ is an unknown error term that depends on $(A, X)$. The assumption is technical and used to prove the error bounds below. The density of the perturbed distribution is $p(\hat{B}, X) = |J_{\hat{b}}(A, X)| u(\hat{b}(A, X))) p(X)$, which may be expanded in $\epsilon$ under the assumption $\epsilon << 1$. Theorem 2.7 characterizes the validity of $C_{\hat{B}}$, i.e. the PIs defined in (9) with $b$ replaced by $\hat{b}$, up to $o(\epsilon^2)$ errors. We assume $b$ and $\hat{b}$ fulfill the requirements of Assumption 2.2, $b$ satisfies the assumption of Corollary 2.6, and $\hat{b}$ is the minimizer of (14) for a given target distribution $U_B$. To simplify the notation, we let $B = b_X(A)$ where $b_X = b(A, X)$ (idem $b_X^{-1}$, $\hat{b}_X$, and $\hat{b}_X^{-1}$) and define $\tilde{B} = \psi_X(A)$, where $\psi_X = b_{X_{N+1}}^{-1} \circ \hat{b}_{X_{N+1}} \circ \hat{b}_X^{-1} \circ b_X$. We bound the validity gap of $C_{\hat{B}}$ in terms of the variation distance between the distributions of $B$ and $\tilde{B}$, i.e.

$$d_{\text{TV}}(P_{BX}, P_{\tilde{B}X}) = \sup_{(A,X)} \|p(B, X) - p(\tilde{B}, X)\| \quad (17)$$

where $p(B, X)$ and $p(\tilde{B}, X)$ are the densities of $P_{BX} = U_B P_X$ and $P_{\tilde{B}X}$ and $B$ and $\tilde{B}$ depend on $(A, X)$ through $b$ and $\psi$. We use the Maximal Coupling Theorem to link the CP validity bound in (2.3) and the total variation distance above. See Lindvall [2002] or Ross and Peköz [2023] for an overview of coupling methods. Up to $o(\epsilon^2)$ corrections, an explicit lower bound of the gap is given in

**Theorem 2.7** *Let* $b(A, X)$ *and* $\hat{b}(A, X)$ *obey Assumption 2.2 and* $U_B = \text{Uniform}([0, 1])$. *Assume* $\hat{b}$ *obeys* (16) *for all* $(A, X)$. *Then,*

$$\text{Prob}(B_{N+1} \leq Q_{\hat{B}}) \quad (18)$$
$$\geq \frac{\lceil (N+1)(1-\alpha) \rceil}{N+1} - \frac{1}{2} d_{\text{TV}}(P_B, P_{\tilde{B}}) \quad (19)$$
$$\geq \frac{\lceil (N+1)(1-\alpha) \rceil}{N+1} - \epsilon \sup_x \|p_X(x)\| L_\delta L_{b^{-1}} + o(\epsilon^2)$$

*where* $Q_{\hat{B}}$ *is the sample quantile of* $\{\hat{b}(A_n, X_n)\}_{n=1}^N$ *defined in* (5) *with $b$ replaced by $\hat{b}$, $\tilde{B} = \psi_X(B)$, $\psi_X = b_{X_{N+1}}^{-1} \circ \hat{b}_{X_{N+1}} \circ \hat{b}_X^{-1} \circ b_X$, $b_X(A) = b(A, X)$ (idem $b_X^{-1}$, $\hat{b}_X$, and $\hat{b}_X^{-1}$), $L_\delta$ and $L_{b^{-1}}$ are the Lipschitz constants of $\delta(B, X)$ and $b^{-1}$, and $p(X)$ is the marginal density of the covariates.*

Theorem 2.7 connects our work with the non-exchangeability gaps obtained in Barber et al. [2022] in a different framework.

## 3 IMPLEMENTATION

We compare two models trained with the proposed scheme, a standard CP algorithm, and the ER model of Papadopoulos et al. [2008]. For simplicity, we focus on Split CP, where the regressor, $f$, is pre-trained on separate data and kept fixed.

### 3.1 DATA

We generate 4 synthetic data sets by perturbing the output of a polynomial regression model of order 2 with four types of heteroscedastic noise. Each data sets consist of 1000 samples of a pair of random variables, $(X, Y)$, obeying

$$Y = X^T w + \epsilon_i, \tag{20}$$
$$X = [1, X_1, X_1^2], \quad X_1 \sim \text{Uniform}([-1, 1]),$$
$$\epsilon_i = 0.1 + \sigma_{\texttt{synth-i}}(X)E, \quad E \sim \mathcal{N}(0, 1)$$

where $w \in \mathbb{R}^3$ is a randomly generated fixed parameter, $\texttt{i} \in \{\texttt{cos}, \texttt{squared}, \texttt{inverse}, \texttt{linear}\}$, and

$$\sigma_{\texttt{synth-cos}}(X) = 2\cos\left(\frac{\pi}{2}X_1\right)\mathbf{1}(X_1 < 0.5) \tag{21}$$

$$\sigma_{\texttt{synth-squared}}(X) = 2X_1^2\mathbf{1}(X_1 > 0.5) \tag{22}$$

$$\sigma_{\texttt{synth-inverse}}(X) = 2\frac{1}{0.1 + |X_1|}\mathbf{1}(X_1 < 0.5) \tag{23}$$

$$\sigma_{\texttt{synth-linear}}(X) = 2|X_1|\mathbf{1}(X_1 > 0.5) \tag{24}$$

For the real-data experiments, we use the following 6 public benchmark data sets from the UCI database: `bike`, the Bike Sharing Data Set [Fanaee-T, 2013], `CASP`, the Physicochemical Properties of Protein Tertiary Structure Data Set [Rana, 2013], `community`, Community and Crime Data Set[Redmond, 2009], `concrete`, the Concrete Compressive Strength Data Set [Yeh, 2007], `energy`, the Energy Efficiency Data Set [Tsanas and Xifara, 2012], and `facebook_1`, the Facebook Comment Volume Data Set Singh [2016].

All data sets are split into two subsets. We use the first subset to train a Random Forest (RF) regressor and the second subset to train and test the conformity functions. For stability, we limit the attribute dimensions to 10 (with PCA) and normalize the label before training the RF models. The Mean Absolute Error of the RF regressor is reported in Table 1. To make the performance comparable across different data sets, we reduce the size of the second subset to 1000 (except for `community`, `concrete`, and `energy` that have size 997, 515, and 384), split it into two equal parts, and use the first to train the conformity measures and the second for calibrating and testing the optimized models.

| data set | MAE |
|---|---|
| synth-cos | 0.051(0.033) |
| synth-inverse | 0.056(0.007) |
| synth-linear | 0.157(0.054) |
| synth-squared | 0.125(0.038) |
| bike | 0.028(0.001) |
| CASP | 0.14(0.003) |
| community | 0.007(0.001) |
| concrete | 0.051(0.002) |
| energy | 0.019(0.001) |
| facebook_1 | 0.003(0.0) |

Table 1: Averages and standard deviation over 5 runs of the MAE of the RF regression model on the synthetic and real data sets.

### 3.2 MODELS

We let $A = |Y - f(X)|$ and consider four model classes,

$$b_{\texttt{baseline}} = A \tag{25}$$

$$b_{\texttt{ER}} = \frac{A}{\gamma + |g(X)|} \tag{26}$$

$$b_{\texttt{Gauss}} = \log\left(\frac{A}{\gamma + |g(X)|}\right) \tag{27}$$

$$b_{\texttt{Uniform}} = \sigma\left(\frac{A}{\gamma + |g(X)|}\right) \tag{28}$$

where $\gamma = 0.001$ and $g$ is a fully connected ReLU neural network with 5 layers of 100 hidden units per layer. The network parameters of ER are trained as in Papadopoulos et al. [2008] by minimizing $\ell_{ER} = \mathrm{E}(|g(X)| - |f(X) - Y|)^2$. Gauss and Uniform are trained with the proposed approach by maximizing the log-likelihood in (14) where $U_B = \mathcal{N}(0, 1)$ for Gauss and $U_B = \text{Uniform}([0, 1])$ for Uniform. The model functional form guarantees $b$ belongs to the distribution support for any $(A, X)$. We use the ADAM gradient descent algorithm of Adam and Lorraine [2019] to solve all optimization problems with standard parameters. The learning rate is 0.01 for ER, $10^{-4}$ for Gauss, and $10^{-5}$ for Uniform.

### 3.3 RESULTS

To evaluate the PIs, we consider their empirical validity, $\mathrm{E}(\mathbf{1}(Y_{N+1} \in C_B))$, average size, $\mathrm{E}(|C_B|)$, and empirical input-conditional coverage, which we approximate with the Worse-Slab Coverage (WSC) algorithm of Cauchois et al. [2020]. Table 2 summarizes our numerical results across the 4 synthetic and 6 real data sets for three values of the confidence level, $1 - \alpha \in \{0.95, 0.90, 0.65\}$. Tables 3 and 4 show the model performances at $\alpha = 0.05$ on each data set. The figures are the averages and standard deviations over 5 random train-test splits.

baseline is the best method for $\alpha = 0.35$, on synthetic and real data, but is generally outperformed by the trained models at higher confidence levels. Gauss seems to outperform all other models on synthetic data. This may be due to the Gaussianity of the noise in the generation of the synthetic samples. On synthetic data, ER is the second best model, probably because we generate the data using $Y \sim f(X) + g(X)\epsilon, \epsilon \sim \mathcal{N}(0,1)$, which implies the ER assumptions are *exact*. Uniform is the best model on real data. Interestingly, the model with the conditional coverage closest to the nominal is not the same at all confidence levels. Table 4 suggests that the optimized models are outperformed by baseline when data are not heteroscedastic, i.e. when baseline has good conditional coverage. This seems to be a shared problem of reweighting methods, as already observed in Romano et al. [2019]. The code for reproducing all numerical simulations is available in this gitHub repository.

# 4  RELATED WORK

**Calibration training** In CP, learning a conformity function from data is fairly new. To the best of our knowledge, the only example of a trained conformity function is the ER algorithm of Papadopoulos et al. [2008, 2011], Lei and Wasserman [2012], where localization is achieved by reweighting $|Y - f(X)|$ with a pre-trained model of the conditional residual, $|g(X)| \approx \mathrm{E}(|Y - f(X)| \, |X)$. Outside CP, there are several examples of calibration optimization for data science applications [Platt et al., 1999, Zadrozny and Elkan, 2002, Naeini et al., 2015]. See Guo et al. [2017] for an introduction and empirical comparison of different calibration methods for neural networks.
**Object-dependent conformity measures.** Papadopoulos et al. [2008, 2011], Lei and Wasserman [2012] use different versions of reweighted conformity measures. The localization function is either fixed, e.g. a KNN-based variance estimator, or pre-trained using *ad-hoc* strategies. Section 5 of Romano et al. [2019] contains a detailed discussion on the limitations of ER. Despite its intuitive and empirical efficiency, ER has been poorly investigated or justified from a theoretical perspective. Our work provides a conceptual framework to explain why it works well for approximating conditional validity [Lei and Wasserman, 2012, Foygel Barber et al., 2021]. Recent work about ER includes Vovk et al. [2020], which is a theoretical study on the validity of oracle conformity measures, and Bellotti [2021], where the conformity score is iteratively updated to make the PI conditionally valid. Similar to Gibbs and Candes [2021], coverage is corrected by minimizing an empirical estimation of the validity gap. Besides Papadopoulos et al. [2008], Bellotti [2020], conformity scores other than $A = |f(X) - Y|$ have been rarely used. In Romano et al. [2019], the conformity function is redesigned to mimic the pinball loss of quantile regression problems. In Colombo

| synthetic data (all data sets) | | | |
|---|---|---|---|
| | coverage | size | WSC |
| $\alpha = 0.05$ | | | |
| baseline | 0.954(0.021) | 0.783(0.086) | 0.798(0.198) |
| ER | 0.954(0.019) | 0.57(0.122) | 0.915(0.094) |
| Gauss | 0.953(0.02) | 0.506(0.048) | 0.883(0.118) |
| Uniform | 0.955(0.025) | 0.624(0.077) | 0.881(0.131) |
| $\alpha = 0.1$ | | | |
| baseline | 0.904(0.035) | 0.546(0.063) | 0.637(0.239) |
| ER | 0.902(0.027) | 0.429(0.067) | 0.833(0.147) |
| Gauss | 0.904(0.024) | 0.382(0.035) | 0.8(0.119) |
| Uniform | 0.906(0.036) | 0.451(0.041) | 0.737(0.153) |
| $\alpha = 0.35$ | | | |
| baseline | 0.661(0.042) | 0.183(0.018) | 0.244(0.146) |
| ER | 0.677(0.052) | 0.209(0.029) | 0.443(0.134) |
| Gauss | 0.673(0.05) | 0.2(0.022) | 0.461(0.139) |
| Uniform | 0.668(0.05) | 0.22(0.029) | 0.449(0.13) |

| real data (all data sets) | | | |
|---|---|---|---|
| | coverage | size | WSC |
| $\alpha = 0.05$ | | | |
| baseline | 0.955(0.016) | 0.141(0.013) | 0.929(0.066) |
| ER | 0.955(0.02) | 0.175(0.056) | 0.94(0.079) |
| Gauss | 0.957(0.013) | 0.17(0.03) | 0.913(0.078) |
| Uniform | 0.952(0.017) | 0.137(0.015) | 0.944(0.076) |
| $\alpha = 0.1$ | | | |
| baseline | 0.901(0.029) | 0.103(0.009) | 0.853(0.134) |
| ER | 0.9(0.032) | 0.117(0.024) | 0.832(0.114) |
| Gauss | 0.911(0.025) | 0.113(0.011) | 0.889(0.093) |
| Uniform | 0.901(0.028) | 0.102(0.009) | 0.869(0.11) |
| $\alpha = 0.35$ | | | |
| baseline | 0.681(0.044) | 0.045(0.005) | 0.512(0.157) |
| ER | 0.659(0.039) | 0.049(0.006) | 0.645(0.143) |
| Gauss | 0.658(0.041) | 0.048(0.003) | 0.574(0.144) |
| Uniform | 0.672(0.054) | 0.046(0.005) | 0.643(0.122) |

Table 2: Average efficiency of the model PIs (coverage, size, and Worst Slab Coverage (WSC) estimate of the conditional coverage [Cauchois et al., 2020]) over the data sets listed in Table 1. The reported averages and standard deviation are computed over 5 random training-test splits.

| synthetic data ($\alpha = 0.05$) | | |
| --- | --- | --- |
| coverage | size | WSC |

| | coverage | size | WSC |
| --- | --- | --- | --- |
| | synth-cos | | |
| baseline | 0.944(0.029) | 0.274(0.039) | 0.749(0.293) |
| ER | 0.945(0.022) | 0.208(0.123) | 0.926(0.062) |
| Gauss | 0.945(0.024) | 0.128(0.015) | 0.833(0.136) |
| Uniform | 0.945(0.035) | 0.148(0.03) | 0.828(0.188) |
| | synth-inverse | | |
| baseline | 0.953(0.02) | 0.321(0.048) | 0.758(0.231) |
| ER | 0.96(0.017) | 0.165(0.019) | 0.942(0.064) |
| Gauss | 0.944(0.024) | 0.124(0.021) | 0.787(0.225) |
| Uniform | 0.951(0.021) | 0.195(0.024) | 0.814(0.233) |
| | synth-linear | | |
| baseline | 0.967(0.018) | 1.923(0.157) | 0.826(0.131) |
| ER | 0.96(0.011) | 1.54(0.309) | 0.941(0.063) |
| Gauss | 0.967(0.008) | 1.402(0.109) | 0.927(0.09) |
| Uniform | 0.968(0.014) | 1.795(0.201) | 0.898(0.071) |
| | synth-squared | | |
| baseline | 0.953(0.016) | 0.614(0.099) | 0.858(0.137) |
| ER | 0.949(0.026) | 0.366(0.037) | 0.849(0.185) |
| Gauss | 0.956(0.024) | 0.37(0.048) | 0.987(0.018) |
| Uniform | 0.956(0.029) | 0.359(0.054) | 0.983(0.03) |

Table 3: Efficiency of the model PIs at $\alpha = 0.05$ on the synthetic data sets listed in Table 1. The reported averages and standard deviation are computed over 5 random training-test splits.

[2023], a series of trained conformity functions are tested empirically. Compared to this work, the learning scheme is not analyzed theoretically and uses a different learning loss. We are unaware of other works where the conformity measure is explicitly optimized. **CP localization and conditional validity.** Except for Papadopoulos et al. [2008], the scheme can be combined with other localization methods because it applies to any base conformity score. Papadopoulos et al. [2008] is an exception because the conformity function is trained by minimizing $\mathbb{E}_{XY}|A^2 - g^2(X)|^2$. In Lei and Wasserman [2014], Vovk [2012], Lin et al. [2021], Guan [2023], Deutschmann et al. [2023], locally adaptive PI are constructed by reweighting the calibration samples and temporarily breaking data exchangeability. The weights transform the marginal distribution into an estimate of the object-conditional distribution. Often, computing the localizing weights requires a density estimation step based on one or more hyper-parameters [Lei and Wasserman, 2014, Vovk, 2012, Guan, 2023, Deutschmann et al., 2023]. This may cause technical issues and can be unreliable if data is scarce. Our approach avoids an explicit estimation, because $b$ is a globally defined functional, and does not require splitting the calibration set. Conditional validity gaps can be viewed as a non-exchangeability problem. Barber et al. [2022] is a study of CP under general non-exchangeability but does not make an explicit connection to local adaptivity. Xu and Xie [2023] exploits the bounds of Barber et al.

| real data ($\alpha = 0.05$) | | |
| --- | --- | --- |
| coverage | size | WSC |

| | coverage | size | WSC |
| --- | --- | --- | --- |
| | bike | | |
| baseline | 0.98(0.011) | 0.112(0.011) | 0.995(0.011) |
| ER | 0.969(0.019) | 0.181(0.095) | 0.995(0.011) |
| Gauss | 0.972(0.007) | 0.127(0.026) | 0.98(0.017) |
| Uniform | 0.976(0.012) | 0.119(0.021) | 0.982(0.02) |
| | CASP | | |
| baseline | 0.944(0.021) | 0.413(0.026) | 0.969(0.041) |
| ER | 0.959(0.011) | 0.435(0.036) | 0.883(0.192) |
| Gauss | 0.973(0.009) | 0.525(0.063) | 0.959(0.045) |
| Uniform | 0.955(0.011) | 0.433(0.03) | 0.89(0.196) |
| | community | | |
| baseline | 0.965(0.011) | 0.059(0.018) | 0.888(0.097) |
| ER | 0.967(0.009) | 0.035(0.016) | 0.967(0.02) |
| Gauss | 0.968(0.011) | 0.071(0.028) | 0.924(0.06) |
| Uniform | 0.963(0.013) | 0.03(0.007) | 0.958(0.052) |
| | concrete | | |
| baseline | 0.96(0.023) | 0.154(0.016) | 0.938(0.064) |
| ER | 0.947(0.031) | 0.288(0.134) | 0.948(0.078) |
| Gauss | 0.95(0.016) | 0.201(0.045) | 0.954(0.043) |
| Uniform | 0.958(0.021) | 0.159(0.014) | 0.972(0.039) |
| | energy | | |
| baseline | 0.942(0.016) | 0.092(0.005) | 0.965(0.046) |
| ER | 0.944(0.041) | 0.098(0.05) | 0.897(0.131) |
| Gauss | 0.944(0.017) | 0.079(0.015) | 0.793(0.18) |
| Uniform | 0.933(0.028) | 0.071(0.019) | 0.975(0.033) |
| | facebook_1 | | |
| baseline | 0.94(0.014) | 0.014(0.003) | 0.82(0.139) |
| ER | 0.947(0.011) | 0.016(0.005) | 0.953(0.045) |
| Gauss | 0.935(0.02) | 0.015(0.004) | 0.87(0.126) |
| Uniform | 0.929(0.015) | 0.01(0.002) | 0.885(0.114) |

Table 4: Efficiency of the model PIs at $\alpha = 0.05$ on the real data sets listed in Table 1. The reported averages and standard deviation are computed over 5 random training-test splits.

[2022] for proving the asymptotic convergence of the estimated PIs to the exact conditional PIs. Theorem 4 in Guan [2023] guarantees exact conditional coverage for a sample reweighting method, up to corrections on the estimated PI. The NF setup allows more explicit bounds on the validity of the algorithm outputs (Theorem 2.7 in Section 2). In Einbinder et al. [2022], a point-prediction model is trained to guarantee $P_{AX} = U_A P_X$, where $U_A = \text{Uniform}([0,1])$. It is unclear whether tuning the point-prediction model or the conformity function produces equivalent PIs. This work is intuitively close to conformity-aware training, which aims to optimize the output of a standard CP algorithm by tuning the underlying model [Colombo and Vovk, 2020, Bellotti, 2020, Stutz et al., 2021, Einbinder et al., 2022]. The two ideas are compatible and could be implemented simultaneously. We leave this for future work.

## 5 DISCUSSION AND LIMITATIONS

This is mainly a theoretical and methodological work. We recognize our numerical simulations are limited, especially regarding the model complexity. We also miss a full comparison with existing localization approaches. We focus on conformity functions similar to ER to underline the efficiency of the learning strategy, without bias coming from the definition of more or less suitable model classes. Generalizing the approach to more complex NF is possible, provided $b(A, X)$ remains invertible, i.e. monotonic in $A$. A comparison with other localization methods goes beyond our scope because calibration training is orthogonal to many existing strategies, e.g. algorithms based on reweighting the calibration samples. The proposed scheme could be used on top of them to provide theoretical guarantees. As mentioned in Section 4, CP-aware retraining of the prediction model could also be combined with calibration training.

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

# 6 PROOFS

**Proof of Lemma 2.1**  . Assume ties occur with probability 0. According to (5), $Q_Z$ is the $n* = \lceil (1-\alpha)(N+1) \rceil$-th smallest element of $\{Z_n\}_{n=1}^N$. Assume the calibration samples have been labeled so that $Z_1 < Z_2 \cdots < Z_{N-1} < Z_N$. By assumption, $Z_1, \ldots, Z_n$, and $Z_{N+1}$ are exchangeable. This implies $Z_{N+1}$ falls with equal probability in any of the $N+1$ intervals

$$(-\infty, Z_1), [Z_1, Z_2) \ldots, [Z_{n*-1}, Q_Z),$$
$$(Q_Z, Z_{n*+1}) \ldots (Z_{N-1}, Z_N), [Z_N, \infty) \quad (29)$$

i.e. $\text{Prob}(Z_{N+1} \leq Q_Z) = \frac{n*}{N+1} = \frac{\lceil (1-\alpha)(N+1) \rceil}{N+1}$. $\square$

**Proof of Lemma 2.3**  $\{B_n\}_{n=1}^N$ are i.i.d. random variable because $b$ is deterministic and $\{A_n\}_{n=1}^N$ are i.i.d. When $b$ satisfies Assumption 2.2, $\text{Prob}(A_n = A_{n'}) = 0$ for any $n \neq n'$ implies $\text{Prob}(B_n = B_{n'}) = \text{Prob}(A_n = b^{-1}(b(A_{n'}, X_{n'}), X_n)) = 0$ for any $n \neq n'$, i.e. there are no ties in $\{B_n\}_{n=1}^N$. Let $Q_B$ be the $(1-\alpha)$-th sample quantile $\{B_n\}_{n=1}^N$ defined in (5). From Lemma 2.1, $\text{Prob}(B_{N+1} \leq Q_B) = \frac{n_*}{N+1}$, with $n_* = \lceil (1-\alpha)(N+1) \rceil$. Let $b_X(A) = b(A, X)$ and $b_X^{-1}(B) = b^{-1}(B, X)$, with $b^{-1}$ defined by $b(b^{-1}(B, X), X) = b_X \circ b_X^{-1}(B)$, and $\partial_A b_X(A) = J_{b\,11}(A, X) = \frac{\partial}{\partial A} b(A, X) = \frac{\partial}{\partial A} b_X(A)$. By Assumption 2.2, $\partial_A b_X > 0$ for all $X$. Let $\frac{d}{ds} h(s, g(s)) = \partial_s h + \partial_g h \partial_s g$ be the total derivative of $h$. From $1 = \frac{d}{dB} b_X \circ b_X^{-1}(B) = \partial_A b_X(b_X^{-1}(B)) \frac{d}{dB} b_X^{-1}(B)$, we obtain $\frac{d}{dB} b_X^{-1}(B) = \left( \partial_A b_X(b_X^{-1}(B)) \right)^{-1} > 0$, i.e. $b_{X_{N+1}}^{-1}(B)$ is a

monotonic function of $B$. Therefore,

$$\text{Prob}(B_{N+1} \leq Q_B) \quad (30)$$
$$= \text{Prob}\left( b_{X_{N+1}}^{-1}(B_{N+1}) \leq b_{X_{N+1}}^{-1}(Q_B) \right) \quad (31)$$
$$= \text{Prob}\left( b_{X_{N+1}}^{-1} \circ b_{X_{N+1}}(A_{N+1}) \leq b_{X_{N+1}}^{-1}(Q_B) \right) \quad (32)$$
$$= \text{Prob}\left( A_{N+1} \leq b_{X_{N+1}}^{-1}(Q_B) \right) \quad (33)$$
$$= \text{Prob}\left( |f(X_{N+1}) - Y_{N+1}| \leq b_{X_{N+1}}^{-1}(Q_B) \right) \quad (34)$$
$$= \text{Prob}(Y_{N+1} \in C_B) \quad (35)$$

where $C_B$ is defined in (7). $\square$

**Proof of Lemma 2.4**  Let $\{B_n = b_{X_n}(A_n)\}_{n=1}^{N+1}$, where $b_X(A) = b(A, X)$, and $C_A$ and $C_B$ be the PIs in (7) and (9). From (5), there are $m_*$ and $n_*$ such that $Q_A = A_{m*}$ and $Q_{\hat{B}} = b_{X_{n_*}}(A_{n*})$. Then, when $b_{X_{N+1}}(A_n) \neq b_{X_n}(A_n)$ for any $n$, we have $b_{X_{N+1}}(A_{n*}) \neq b_{X_{n_*}}(A_{n*})$ and

$$|C_B| = b_{X_{N+1}}^{-1} \circ b_{X_{m*}}(A_{m*}) \neq A_{n*} = |C_A| \quad (36)$$

The claim holds because $A_{n*} = b_{X_{N+1}}^{-1} \circ b_{X_{m*}}(A_{m*})$ occurs with probability 0 if $A_n$ are continuous. $\square$

**Proof of Theorem2.5**  Let $\{A_n \sim P_A\}_{n=1}^N \{\tilde{A}_n \sim P_{A|X_{N+1}}\}_{n=1}^N$ be two collections of i.i.d random variables distributed according to the marginal and $X_{N+1}$-conditional distributions. Let $Q_A$ and $Q_{\tilde{A}}$ be the sample quantiles of the two collections defined in (5). Let $C_A$ be the PI defined in (7) and $C_{\tilde{A}}$ be obtained analogously with $Q_A$ replaced by $Q_{\tilde{A}}$. Assume ties occur with probability 0. By the Bayesian theorem, $P_{AX} = P_A P_X$ implies $P_{A|X} = P_A = \sum_X P_{AX}$. Then, for any $X_{N+1}$, $\tilde{A}_n \sim P_A \sim A_n$ and, from Lemma 2.1, $\text{Prob}(\tilde{A}_{N+1} \leq Q_{\tilde{A}}) = \text{Prob}(\tilde{A}_{N+1} \leq Q_A) = \text{Prob}(Y_{N+1} \in C_A)$. $\square$

**Proof of Corollary 2.6**  Let $Q_{A|X_{N+1}}$ and $C_{A|X_{N+1}}$ be the conditional sample quantile of $\{\tilde{A}_n \sim P_{A|X_{N+1}}\}$ and the corresponding PI defined as in (7) with $Q_A$ replaced by $Q_{A|X_{N+1}}$. By construction, $C_{A|X_{N+1}}$ is conditionally valid at $X_{N+1}$, i.e. it obeys $\text{Prob}(Y_{N+1} \in C_{A|X_{N+1}} | X_{N+1}) = \frac{m_*}{N+1}$, $m_* = \lceil (1-\alpha)(N+1) \rceil$. Let $(B, X) = (b(A, X), X) = (b_X(A), X)$. Then, if $b$ obeys Assumption 2.2 and $P_{BX} = P_{B|X} P_X = U_B P_X$,

$$Q_{A|X_{N+1}} = Q_{b_{X_{N+1}}^{-1}(B)|X_{N+1}} \quad (37)$$
$$= b_{X_{N+1}}^{-1}(Q_{B|X_{N+1}}) = b_{X_{N+1}}^{-1}(Q_B) \quad (38)$$

because $b_{X_{N+1}}^{-1}$ is monotonic and we apply it globally to all samples (second equality) and $P_{BX} = U_B P_X$ implies $Q_{B|X_{N+1}} = Q_B$ (last equality). The claim follows from Lemma 2.1, $b_{X_{N+1}}(A) = b(A, X_{N+1})$, and the PI definition in (9). $\square$

**Proof of Theorem 2.7.** Let $\{A_n\}_{n=1}^{N+1}$ be a collection of i.i.d. conformity scores and $\{B_n = b(A_n, X_n)\}_{n=1}^{N+1}$ and $\{\hat{B}_n = \hat{b}(A_n, X_n)\}_{n=1}^{N+1}$ the conformity scores transformed by $b$ and $\hat{b} = (1 - \epsilon)b + \epsilon\delta$. Let $C_B$ be the PI defined in (9) and $C_{\hat{B}}$ defined analogously by replacing $b$ with $\hat{b}$. Let $b_X(B) = b(A, X)$ (idem $\hat{b}_X$, $b_X^{-1}$, and $\hat{b}_X^{-1}$). Assumption 2.2 and Corollary 2.6 imply

$$\text{Prob}(Y_{N+1} \in C_{\hat{B}} | X_{N+1}) \tag{39}$$

$$= \text{Prob}(A_{N+1} \leq \hat{b}_{X_{N+1}}^{-1}(Q_{\hat{B}}) | X_{N+1}) \tag{40}$$

$$= \text{Prob}(b_{X_{N+1}}^{-1}(B_{N+1}) \leq \hat{b}_{X_{N+1}}^{-1}(Q_{\hat{B}}) | X_{N+1}) \tag{41}$$

$$= \text{Prob}(B_{N+1} \leq b_{X_{N+1}} \circ \hat{b}_{X_{N+1}}^{-1}(Q_{\hat{B}})) \tag{42}$$

where we drop the conditioning in the last line because, by assumption, $(B_n, X_n) \sim U_B P_X$ for all $X_n$. The monotonicity of $b_{X_{N+1}} \circ \hat{b}_{X_{N+1}}^{-1}(B)$ implies $\hat{b}_{X_{N+1}}^{-1}(Q_{\hat{B}}) = Q_{\tilde{B}}$, where $Q_{\tilde{B}}$ is the sample quantile of $\{\tilde{B}_n\}$, $\tilde{B}_n = b_{X_{N+1}} \circ \hat{b}_{X_{N+1}}^{-1}(\hat{B}_n)$. Test and calibration data are not exchangeable because, $B_{N+1}$ and $\tilde{B}_n$, $n = 1, \ldots N$, come from different distributions. The coverage gap can be bounded in terms of the total variation distance between their distributions, $P_B$ and $P_{\tilde{B}}$, i.e. $\text{d}_{\text{TV}}(P_B, P_{\tilde{B}}) = \sup_Z |P_B(Z) - P_{\tilde{B}}(Z)|$. Let $(\hat{P}, \tilde{B}, B')$ define a *maximal coupling* between $\tilde{B}_1, \ldots, \tilde{B}_N$ and $B_{N+1}$ defined by $\text{Prob}(\tilde{B}_n) = \hat{P}(\tilde{B})$, $n = 1, \ldots, N$, and $\text{Prob}(B_{N+1}) = \hat{P}(B')$. Then,

$$\text{Prob}(B_{N+1} \leq Q_{\tilde{B}}) \tag{43}$$

$$= \hat{P}(B' \leq Q_{\tilde{B}}, B' = \tilde{B}) + \hat{P}(B' \leq Q_{\tilde{B}}, B' \neq \tilde{B}) \tag{44}$$

$$\geq \frac{\lceil (N+1)(1-\alpha) \rceil}{N+1} - \hat{P}(B' \neq \tilde{B})) \tag{45}$$

where the Maximal Coupling Theorem implies (see Lindvall [2002], Ross and Peköz [2023] for a proof)

$$\hat{P}(B' \neq \tilde{B}) = \frac{1}{2}\text{d}_{\text{TV}}(P_{B_{N+1}}, P_{\tilde{B}_n}) \tag{46}$$

which, in this case, holds for any $n \in \{1, \ldots, N\}$ because we assume the data objects are i.i.d.

Assume $\hat{b} = (1 - \epsilon)b + \epsilon\delta$ and $\hat{b}^{-1} = (1 - \epsilon)b^{-1} + \epsilon\delta^{-1}$, for all $(A, X)$. The invertibility of $\hat{b}$ implies $\text{Id} = \hat{b} \circ \hat{b}^{-1} = (1 - 2\epsilon)\text{Id} + \epsilon(b \circ \delta^{-1} + b^{-1} \circ \delta) + \epsilon^2(\text{Id} + \delta \circ \delta^{-1})$, where $\text{Id}(B) = B$. Neglecting second-order terms, we have $b \circ \delta^{-1} + \delta \circ b^{-1} = 2\text{Id}$, i.e. $\delta^{-1} = 2b^{-1} - b^{-1} \circ \delta \circ b^{-1}$ and $\hat{b}^{-1} = (1 - \epsilon)b^{-1} + \epsilon(2b^{-1} - b^{-1} \circ \delta \circ b^{-1}) = (1 + \epsilon)b^{-1} - b^{-1} \circ \delta \circ b^{-1}$. Let $b_X(A) = b(A, X)$ (idem $b^{-1}$, $\hat{b}$, $\hat{b}^{-1}$, and $\delta$). Since $\psi_X(B) = b_{X_{N+1}} \circ \hat{b}_{X_{N+1}}^{-1} \circ \hat{b}_X \circ b_X^{-1}$ is monotonic, we may interpret it as an NF. The density of $(\tilde{B}_n, X_n) \sim P_{\tilde{B}X}$ is

$$p(\psi_X(B), X) = \frac{p(B, X)}{|\det J_\psi(B, X)|} = \frac{u(B)p(X)}{|\partial_B \psi_X(B)|} \tag{47}$$

where $|\partial_B \psi_X(B)| = \partial_B \psi_X(B)$ because $\psi_X$ is monotonic. Then, up to $o(\epsilon^2)$ errors,

$$\text{d}_{\text{TV}}(P_B, P_{\tilde{B}}) \tag{48}$$

$$= \sup_{(B,X)} \left\| u(B)p(B)\left(1 - \frac{1}{\partial_B \psi_X(B)}\right) \right\| \tag{49}$$

$$\leq \epsilon \sup_{(B,X)} \|u(B)p(X)\| \sup_{(B,X)} \|1 - \partial_B \psi_X^{-1}(B)\| \tag{50}$$

$$= \epsilon \sup_{(B,X)} \|u(B)p(X)\| \tag{51}$$

$$\times \sup_{(B,X)} \|\partial_A \delta_X \circ \partial_B b_X^{-1} - \partial_A \delta_{X_{N+1}} \circ \partial_B b_{X_{N+1}}^{-1}\|$$

$$\leq 2\epsilon \sup_{(B,x)} \|u(B)p(X)\| L_\delta L_{b^{-1}} \tag{52}$$

where $L_\delta$ and $L_{b^{-1}}$ are the Lipshitz constants of $\delta_X$ and $b_X^{-1}$. If $U_B = \text{Uniform}([0, 1])$, $\sup_{(B,X)} \|u(B)p(X)\| = \sup_X \|p(X)\|$, which only depends on the marginal density of the covariates over the attribute space. Hence,

$$\text{Prob}(B_{N+1} \leq Q_{\hat{B}}) \tag{53}$$

$$\geq \frac{\lceil (N+1)(1-\alpha) \rceil}{N+1} - \epsilon \sup_X \|p(X)\| L_\delta L_{b^{-1}} \tag{54}$$

$\square$