# OpenReview forum: "Normalizing Flows for Conformal Regression"
_auai.org/UAI/2024/Conference — UAI 2024 oral_

### Official Review · Reviewer_ouVS · 2024-03-22

**Q2-1 Originality-Novelty:** 3
**Q2-2 Correctness-Technical Quality:** 3
**Q2-5 Clarity Of Writing:** 3

**Q1 Summary And Contributions:**

This paper presents advances in normalizing flows, aiming to address conformal regression.

**Q2-3 Extent To Which Claims Are Supported By Evidence:**

2: Fair: the main claims are somewhat supported by evidence (but the experimental evaluation may be weak, or does not match entirely with the claims, important baselines may be missing, proofs contain important ideas but lack rigor, algorithmic details are only discussed superficially, references are imprecise, assumptions are not sufficiently motivated or explicated, etc.).

**Q2-4 Reproducibility:**

2: Fair: key resources (e.g. proofs, code, data) are unavailable but key details (e.g. proof sketches, experimental setup) are sufficiently well-described for an expert to confidently reproduce the main results.

**Q3 Main Strengths:**

The paper addresses a hot topic, which is normalizing flows

**Q4 Main Weakness:**

The contributions of this paper are not clear, and not clearly presented.

It is not clear why the authors did not consider the literature de Normalizing flows for (Bayesian) calibrations, such as
Yamauchi Y, Buskirk L, Giuliani P, Godbey K. Normalizing Flows for Bayesian Posteriors: Reproducibility and Deployment. arXiv preprint arXiv:2310.04635. 2023 Oct 7.
Ardizzone L, Mackowiak R, Rother C, Köthe U. Training normalizing flows with the information bottleneck for competitive generative classification. Advances in Neural Information Processing Systems. 2020;33:7828-40.

**Q5 Detailed Comments To The Authors:**

See weaknesses from main comments.

Moreover, there are some sentences that need to be rewritten because missing many things, such as “An less tight additive boun \delta=…”

There are some spelling and grammatical errors, such as: from an unknow joint distribution, An less tight additive boun, LIMITIATIONS

**Q9 Complying With Reviewing Instructions:**

Yes

---

> ### Author Rebuttal · Authors · 2024-04-05
>
> >The contributions of this paper are not clear, and not clearly presented.
>
> We agree that the presentation style is often too technical and may be improved. The novelty is in the *calibration training* idea. Standard Conformal Prediction (CP) algorithms are fixed computational tools applied on top of pre-trained models. Introducing NF to CP and enforcing the distribution of the conformity scores to factorise is also new.
>
> > *It is not clear why the authors did not consider the literature on Normalizing flows for (Bayesian) calibrations, such as Yamauchi Y, Buskirk L, Giuliani P, Godbey K. Normalizing Flows for Bayesian: Reproducibility and Deployment and Ardizzone L, Mackowiak R, Rother C, Köthe U. Training normalizing flows with the information bottleneck for competitive generative classification.*
>
> Thank you for your suggestions.
> - We did not know about Yamauchi et al. CP is normally seen as a (frequentist) alternative to Bayesian approaches. We have probably missed many other Bayesian-related references and are sorry for this. Compared to Adapting Yamamuchi et al.'s methods, our optimization does not require MCMC sampling, which is often computationally inefficient.
> - Many thanks for bringing Adizzone et al.'s paper to our attention. Using their *Information Bottleneck* approach to train the NF in our method would be interesting but requires some work, as they focus on generative classification and we consider standard regression tasks. Is the approach applicable even without the generator step? More generally, the literature on NF is vast, especially in Physics. We only cite [1] in our paper because it presents NF for ML application in its minimal terms.
>
> >*Moreover, there are some sentences that need to be rewritten because missing many things, such as “An less tight additive boun \delta=…” * and *There are some spelling and grammatical errors, such as: from an unknow joint distribution, An less tight additive boun, LIMITIATIONS*
>
> We are sorry about this and all other typos.
>
>
> [1]
> George Papamakarios, Eric Nalisnick, Danilo Jimenez
> Rezende, Shakir Mohamed, and Balaji Lakshminarayanan. Normalizing flows for probabilistic modelling
> and inference. The Journal of Machine Learning Research, 22(1):2617–2680, 2021.

---

### Official Review · Reviewer_5fu9 · 2024-03-23

**Q2-1 Originality-Novelty:** 4
**Q2-2 Correctness-Technical Quality:** 3
**Q2-5 Clarity Of Writing:** 2

**Q1 Summary And Contributions:**

The authors propose a strategy to adapt prediction intervals obtained by CP by training a flow to target the joint distribution of the prediction errors (i.e. standard conformal score in regression) and the inputs. They show validity of obtained PIs.

**Q2-3 Extent To Which Claims Are Supported By Evidence:**

3: Good: the main claims are supported by convincing evidence (in the form of adequate experimental evaluation, proofs, (pseudo-)code, references, assumptions).

**Q2-4 Reproducibility:**

3: Good: key resources (e.g. proofs, code, data) are available and key details (e.g. proofs, experimental setup) are sufficiently well-described for competent researchers to confidently reproduce the main results.

**Q3 Main Strengths:**

The paper addresses a key challenge of excessively wide PIs produced by CP, and provides a theoretically sound method for obtaining adaptive and valid PIs. The idea of using NFs to optimising/learning conformal scores is elegant.

**Q4 Main Weakness:**

Paper is partly difficult to follow. For example, sometimes notation are used on the fly, and then introduced few pages later (e.g. u_Φ in Equation 3). Perhaps it would have been easier to follow the paper if things had been presented in their full generality from the beginning, along with the notations. Moreover, terminology is not always evident. For example: "Normalizing Flow (NF), i.e. a coordinate transformation that maps a target distribution, P , into a target distribution, P′", isn't P is called a base distribution in NF setting. Now, throughout the paper both P and P' are called target distributions, so using word "target" just confuses.

Why would someone want to stick with a specific transformation mentioned in Equation 2? For the reader's convenience, the sentence,"Papadopoulos et al. [2008] is an exception because the conformity function is trained by minimizing $E_{XY} |A^2 − g^2(X)|^2$. In Section 3, we show that training $g(X)$ as an NF may produce better PIs on real-world data.", could appear earlier

How large calibration set should be so that one can consider fitting a flow to a space with the dimensionality of $dim(\mathcal{X}) + 1$? Does one needs increasingly larger calibration sets to get reliable PIs? NFs ability to extrapolate beyond the observed data can be limited, what implications this have on your proposed approach?

Lot of typos even in math.

**Q5 Detailed Comments To The Authors:**

ΦERExp = {φX (A) = Ae−(γ + g(X)2)}... what e? exp(1) or bracket in wrong place?

Italicize the word 'marginal' the first time you use it in the way it is understood throughout the paper: "They are marginal PIs because coverage is defined in terms of the data joint distribution Prob(Y ∈ CX) = PXY X1Y1...XN YN (Y ∈ C), without conditioning on either the object label or X [Vovk, 2012]." I don't think that 'marginal PI' is an established term.

In Figure 1 there reads C_LR but it should be C_ER

Typo: {(An = |f(Xn) − Yn|,Xn)}N+1n=1 =.

Typo. Table 1; basline

**Q9 Complying With Reviewing Instructions:**

Yes

---

> ### Author Rebuttal · Authors · 2024-04-05
>
> Thank you for reading and commenting on our work.
>
> >"Paper is partly difficult to follow. For example, sometimes notations are used on the fly, and then introduced few pages later (e.g. u_Φ in Equation 3). Perhaps it would have been easier to follow the paper if things had been presented in their full generality from the beginning, along with the notations. Moreover, terminology is not always evident. For example: "Normalizing Flow (NF), i.e. a coordinate transformation that maps a target distribution, P , into a target distribution, P′", isn't P is called a base distribution in NF setting. Now, throughout the paper both P and P' are called target distributions, so using word "target" just confuses."
>
> Yes, the notation can be made more consistent and clear throughout the paper. We are sorry about this. We will work on improving the presentation and implement your suggestions if the paper is accepted.
>
> >"Why would someone want to stick with a specific transformation mentioned in Equation 2?
>
> The specific transformation is just an example. In the experiments, we use more general formats. As noted by another reviewer, choosing the right transformation functional may be hard and often requires finding the right trade-off between flexibility and robustness.
>
>
> >"For the reader's convenience, the sentence, "Papadopoulos et al. [2008] is an exception because the conformity function is trained by minimizing $E|A^2-g(X)^2|^2$. In Section 3, we show that training  $g(X)$ as an NF may produce better PIs on real-world data.", could appear earlier."
>
> Thank you for the suggestion.
>
> >"How large calibration set should be so that one can consider fitting a flow to a space with the dimensionality of $dim(X) + 1$? Does one needs increasingly larger calibration sets to get reliable PIs? NFs ability to extrapolate beyond the observed data can be limited, what implications this have on your proposed approach?"
>
> As mentioned above and in the rebuttal to another reviewer, we do not claim to solve the hard problem of finding the right trade-off between flexibility and robustness. Compared to learning more general NFs, however, we only need to train the *all-to-one* transformation $B = f_0(A, X_1, \dots, X_d)$ and leave $X$ unchanged. This implies the Jacobian has only one nontrivial row, $[\partial_A f_0, \nabla f_0]$. In our experiments, we trained the NF on the same set used for the underlying regressor. Varying its side would be an interesting ablation study.
>
> > "Lot of typos even in math.",   "Typo: ${(An = |f(Xn) − Yn|,Xn)}N+1n=1 =$" and "Typo. Table 1; basline"
>
> Sorry about all these typos and thank you for spotting them.
>
> > "exp(1) or bracket in wrong place?"
>
> It should be $A e^{-(\gamma + g(X)^2)}$. Thank you!
>
> >"Italicize the word 'marginal' the first time you use it [...]"
>
> Good suggestion. *Marginal PI* is a shortcut for *Marginally Valid PI*. But indeed, we need to introduce the phrase before using it.
>
> >"In Figure 1 there reads $C_{LR}$ but it should be $C_{ER}$"
>
> Right.

---

### Official Review · Reviewer_3Fh9 · 2024-03-24

**Q2-1 Originality-Novelty:** 3
**Q2-2 Correctness-Technical Quality:** 3
**Q2-5 Clarity Of Writing:** 3

**Q1 Summary And Contributions:**

The paper tackles the problem of adjusting conformal prediction intervals to reflect differences in accuracy between different parts of the input space. Standard calibration procedures in conformal prediction ignore differences between regions of the input space, which leads to suboptimal prediction intervals (often too conservative, and valid only on average across the whole data set). Ideally, we would like to have prediction intervals that are adjusted to different levels of accuracy at each data point. As the authors note, this is impossible in the strict sense (impossibility results exist), but more obtaining better adaptive prediction intervals is highly desirable in practice and interesting in theory. The authors describe a method for jointly learning a family of calibration functions to achieve this goal. This can be seen as training a normalizing flow mapping the distribution of standard conformity scores onto the uniform distribution. The mapped scores are then calibrated in the standard way and the mapping is inverted to obtain adaptive prediction intervals. The authors present theoretical results that attempt to quantify the difference between their proposed intervals and ideal conditionally valid intervals as a function of the quality of fit of the normalizing flow. The experiments show the method generally outperforms standard conformal prediction, and offers some improvement over existing techniques for adaptive intervals.

**Q2-3 Extent To Which Claims Are Supported By Evidence:**

3: Good: the main claims are supported by convincing evidence (in the form of adequate experimental evaluation, proofs, (pseudo-)code, references, assumptions).

**Q2-4 Reproducibility:**

3: Good: key resources (e.g. proofs, code, data) are available and key details (e.g. proofs, experimental setup) are sufficiently well-described for competent researchers to confidently reproduce the main results.

**Q3 Main Strengths:**

The paper addresses an important and timely problem and presents a formal framework for quantifying the discrepancy between the ideal PIs and the PIs obtained by the method. The method is conceptually quite simple, appears to be novel and practically applicable. The core ideas are explained clearly.

**Q4 Main Weakness:**

While the overall approach is clearly explained, the paper is at times fuzzy about the key details and reasoning. Most importantly, I am confused about how the data is split into different sets. I am fairly confident that the method makes sense when the NF is trained on some set S_1, then calibrated on some set S_2. The complete train/test setup would thus involve 4 sets: S_0 for training the underlying model, S_1 for training the NF, S_2 for calibrating the NF, and S_3 for testing. In some places, the authors appear to suggest that fewer sets are necessary like on in Section 2.2 (where it seems it is trained on all of A) or page 2 when they say "data exchangeability is not broken because the transformation does not depend on the test object" except if the NF was trained on the same data points that are used for calibration, the calibration points would not be independent of the NF's parameters so exchangeability with the test point would be broken. This has to be clarified for the method to be understandable.

The authors also say "we avoid overfitting by imposing a smooth functional dependence of \Psi on X and A", but then proceed to use a neural network with many layers on relatively small data sets, which makes the above statement confusing and a bit vacuous.

**Q5 Detailed Comments To The Authors:**

- p.4 under the figure "...random variables {(A_n...}=" - dangling equality sign
- p.2 left panel - instead of using "source" and "target distibution", "target" is used repeatedly, which is confusing
- I am not sure what the difference is between the exp versions of the method and the non-exp versions
- when using a coupling argument and the total variation distance, it has to be more clearly explained that the probability of B not equal to B hat is under a maximal coupling - it is confusing at first when one looks at the equation.

**Q9 Complying With Reviewing Instructions:**

Yes

---

> ### Author Rebuttal · Authors · 2024-04-05
>
> Thank you for your nice review.
>
> >*While the overall approach is clearly explained, the paper is at times fuzzy about the key details and reasoning.*
>
> We agree that the presentation can be improved. We will work on that if the paper is accepted.
>
> >*Most importantly, I am confused about how the data is split into different sets. I am fairly confident that the method makes sense when the NF is trained on some set S_1, then calibrated on some set S_2. The complete train/test setup would thus involve 4 sets: S_0 for training the underlying model, S_1 for training the NF, S_2 for calibrating the NF, and S_3 for testing. In some places, the authors appear to suggest that fewer sets are necessary like on in Section 2.2 (where it seems it is trained on all of A) or page 2 when they say "data exchangeability is not broken because the transformation does not depend on the test object" except if the NF was trained on the same data points that are used for calibration, the calibration points would not be independent of the NF's parameters so exchangeability with the test point would be broken. This has to be clarified for the method to be understandable.*
>
> Right. Ideally, 4 sets are needed, for training the regression model and the NF, calibrating, and testing. To guarantee exchangeability we only need to separate training, calibration and testing. If data are scarce, we can train the NF on the same set used for training the model as in the paper. This would cause the NF to underperform on the calibration and test sets but does not break exchangeability.
>
> >*The authors also say, "we avoid overfitting by imposing a smooth functional dependence of \Psi on X and A", but then proceed to use a neural network with many layers on relatively small data sets, which makes the above statement confusing and a bit vacuous.*
>
> Choosing the right model is not easy. The size of the neural net is comparable with the models used in other approaches for localizing CP, e.g. [1]. Expressive models are needed for achieving $P_{BX} = U_B P_X$, which may be hard on a general data set. In our implementation, we avoid overfitting by $L_2$-regularizing the objective function used to train the NF.
>
> > *p.4 under the figure "...random variables {(A_n...}=" - dangling equality sign*
>
> Thank you!
>
> ->*p.2 left panel - instead of using "source" and "target distribution", "target" is used repeatedly, which is confusing*
>
> We are sorry for the typo.
>
> >*I  am not sure what the difference is between the exp versions of the method and the non-exp versions.*
>
> It is known that monotonic transformations of the conformity score do not affect the interval. We are unsure about the monotonic transformations of the reweighting factor. Empirically, exponentiation does not make a great difference.
>
> >*when using a coupling argument and the total variation distance, it has to be more clearly explained that the probability of B not equal to B hat is under a maximal coupling - it is confusing at first when one looks at the equation.*
>
> Thank you for the remark.
>
>
> [1]
> Romano, Yaniv, Evan Patterson, and Emmanuel Candes. "Conformalized quantile regression." Advances in neural information processing systems 32 (2019).

---

### Meta-Review · Area_Chair_W1jm · 2024-04-16

The paper brings together in a convincing way two trendy topics that are normalizing flows and conformal prediction. All reviewers recommended either accept or strong accept.